# Association of Reallocating Time in Different Intensities of Physical Activity with Weight Status Changes among Normal-Weight Chinese Children: A National Prospective Study

**DOI:** 10.3390/ijerph17165761

**Published:** 2020-08-10

**Authors:** Kaiyun Tan, Li Cai, Lijuan Lai, Zhaohuan Gui, Xia Zeng, Yajie Lv, Jingshu Zhang, Hui Wang, Yinghua Ma, Yajun Chen

**Affiliations:** 1Department of Maternal and Child Health, School of Public Health, Sun Yat-sen University, Guangzhou 510080, China; tanky3@mail2.sysu.edu.cn (K.T.); caili5@mail.sysu.edu.cn (L.C.); lailj@mail2.sysu.edu.cn (L.L.); guizhh@mail2.sysu.edu.cn (Z.G.); zengx26@mail2.sysu.edu.cn (X.Z.); lvyj3@mail2.sysu.edu.cn (Y.L.); zhangjsh29@mail2.sysu.edu.cn (J.Z.); wangh575@mail2.sysu.edu.cn (H.W.); 2Institute of Child and Adolescent Health, School of Public Health, Peking University, Beijing 100191, China

**Keywords:** isotemporal substitution, physical activity, sedentary behavior, obesity, children

## Abstract

*Background:* Time spent in different intensity-specific physical activities is codependent, but the substitution effect of different activities on weight status changes in children remains unclear. This study aims to investigate the prospective association between reallocating time in different intensities of physical activity and weight status changes among Chinese children. *Methods:* A national sample of 15,100 normal-weight children aged 7–18 years (46.7% boys) were recruited in September 2013 and followed up for nine months. Vigorous-intensity physical activity (VPA), moderate-intensity physical activity (MPA), walking, and sedentary time were obtained by International Physical Activity Questionnaire Short Form (IPAQ-SF). Height and weight were objectively measured, by which body mass index (BMI) and BMI z-score were calculated. Weight status was classified by the Chinese criteria for 7- to 18-year-old children. Isotemporal substitution analyses (including single-factor model, partition model, and isotemporal substitution model) were applied to examine the association of time allocation with weight status changes. *Results:* Each 30 min/day of increase in VPA was favorably associated with a 13.2% reduced risk of incident overweight/obesity in a single-factor model and a 15.6% reduced risk in a partition model. Negative associations were found between VPA, MPA, walking and the risk of being underweight in the single-factor model, but not in the partition model. In substitution models, replacing 30 min/day sedentary time with an equal amount of VPA was favorably associated with a 16.1% reduction of the risk of being overweight/obese. *Conclusion:* These findings highlight the need for promoting vigorous-intensity physical activity in children.

## 1. Introduction

Obesity is a global major health problem affecting both adults and children, and it can exert detrimental effects across the life-course. Over 340 million children and adolescents are overweight and obese around the world [1]. In China, 20.5% of children aged 7–18 years were classified as overweight or obese in 2014 [2]. From cross-sectional studies, it has been acknowledged that an inverse association exists between moderate- to vigorous-intensity physical activity (MVPA) and childhood obesity [3]. Accordingly, global guidelines, including from World Health Organization and Department of Health and Human Services in America, recommended at least 60 min per day of MVPA and at least three times per week of vigorous-intensity physical activity (VPA) for children and adolescents to achieve health benefits, including healthy weight status [4,5]. Unfortunately, only 44.1% of children adhere to the MVPA guideline reported in a 12-country study [6], and the lowest adherence was found in China (15.1%). Sedentary behavior represents the lowest intensity spectrum of physical activity and data reported that Chinese children spend approximately 2.9 h per day in leisure-time sedentary [7].

In a finite 24-h day, awake time comprises different intensity-specific physical activities (VPA, moderate-intensity physical activity (MPA), light-intensity physical activity (LPA), and sedentary time), and time allocated to one intensity-specific activity cannot be spent in another. Hence, research should not merely investigate the associations of each intensity behavior in isolation with health outcomes (e.g., cardiovascular health, waist circumference change and weight status), but should consider the substitution of the remaining codependent behavior [8]. Examining this substitution association may help inform evidence-based interventions of replacing sedentary time with ambulatory activity behaviors [9].

Recently, studies have used isotemporal substitution analysis [10] to model the effects of replacing one intensity-specific activity behavior with another in a fixed duration. In adults, several cross-sectional studies have reported that reallocating time of MVPA or LPA into sedentary time was associated with several health outcomes [11,12,13]. However, whether these substitution observations are applicable to children is less clear. At present, replacing sedentary time with MVPA has been deemed beneficial for child obesity [14], but additional prospective studies are needed, particularly on incident childhood obesity. Different from the prevalence, the incidence of being overweight/obese, which is little known [15,16], could provide a deeper insight into the development of obesity and windows of prevention [17]. However, the substitution association between physical activity and incident overweight/obesity was seldom investigated. Understanding this association is crucial for more precise guidelines for reducing sedentary time in children and adolescents.

Therefore, using a national sample of Chinese children aged 7–18 years, the purpose of the current study was to investigate the prospective association of VPA, MPA, walking and sedentary time with the changes in weight status, by applying isotemporal substitution regression analysis.

## 2. Methods

### 2.1. Study Design and Participants

This prospective study was drawn from a national school-based healthy lifestyle intervention program, which was conducted from September 2013 to June 2014 in China. As a multicenter randomized study, this study was undertaken with a multistage cluster sampling method, selecting 94 elementary, middle and high schools from seven provinces/regions: Hunan (central China), Shanghai (eastern China), Guangdong (southern China), Chongqing (western China), Ningxia (northwestern China), Tianjin (northern China), and Liaoning (northeastern China). About 12 to 16 schools were randomly selected from each region. All the students from the school except the highest grade were included. Additional details of the study design and methodology were reported elsewhere [18]. Ethics approval was obtained from the Ethical Committee of Peking University, and written informed consent was sought from participants’ parents.

In the present study, we only included normal-weight students (*n* = 18,302) aged 7–18 years from the control group at baseline. These children were enrolled in baseline measurement and 98.9% of them (*n* = 18,108) participated the follow-up data collection. A total of 3008 students were excluded because of missing data on valid assessment of physical activity. Finally, 15,100 participants were finally enrolled in this analysis. (See Appendix A)

### 2.2. Anthropometric Measurement

Anthropometric variables are measured by professional nurses and doctors according to standard procedures. Standing height and body weight were objectively measured barefoot and with light clothes by trained assessors both at baseline and follow-up. The height was measured to the nearest 0.5 cm, and the weight was measured to the nearest 0.1 kg. Body mass index (BMI) was calculated as weight in kilograms divided by the square of height in meters (kg/m^2^). BMI z-score were calculated according to the age- and sex-specific standards of the World Health Organization [19]. The weight status was classified into underweight, normal weight, and overweight/obesity by using the Chinese criteria [20] for 7- to 18-year-old children. The changes in variables were calculated as follow-up minus baseline.

### 2.3. Questionnaire

A self-reported questionnaire, including children-reported and parent-reported parts, was designed to obtain information both at baseline and follow-up. Trained investigators delivered the questionnaire to children in class and interpreted in detail. Children were required to deliver the questionnaire to their parents. Parents were asked to finish the parent-reported questionnaire and assist their children to fill out the child-reported questionnaire. Both the parent- and child-questionnaire of children in grades 1–3 were reported by parents. After all the answered questionnaires were handed in within three days, the trained investigators would perform quality control by checking integrity and correctness of completion and replenish the missing information by calling their parents.

### 2.4. Physical Activity and Sedentary Behavior

Physical activity was estimated by the International Physical Activity Questionnaire Short Form (IPAQ-SF) during the last seven days. Participants were asked to report the daily average sedentary time (sitting or lying at school and home, but not sleeping) and weekly frequency and duration of following activities: VPA (activities that cause people to be out of breath, perspire and experience extreme exhaustion, such as running, playing basketball and football, swimming, doing aerobics, or carrying a heavy load), MPA (activities that cause people mildly perspire and experience slight exhaustion, such as bicycling, playing table tennis and badminton, dancing, but not walking), and walking (including at school, at home, for transport, and for exercise). The total time was coded as the sum of time spent in sedentary, walking, MPA, and VPA. Acceptable reliability (coefficient: PA, 0.46; sedentary, 0.59) and validity (coefficient: PA, 0.43; sedentary, 0.60) of the questionnaires of physical activity were assessed among a sample of 298 children.

### 2.5. Other Covariates

Covariates were obtained from the questionnaire, including participants’ age, sex, parental educational levels, urban/rural residence, family history of obesity, sleep duration and diet intake. Parental educational levels were defined as the highest level of either parents (junior high school or below, senior high school, junior college, and college or above). Sleep duration was defined as 9–11 h or not. Diet intake was defined as the consumption of sugary beverages (yes or no), high-energy snacks (yes or no), and fried food (yes or no) during the last seven days.

### 2.6. Statistical Analysis

Descriptive results were presented as means (standard deviation) for continuous variables and frequency (proportions) for categorical variables. Student’s *t*-tests and chi-square tests were used to evaluate sex differences for continuous variables and categorical variables, respectively. Children were stratified into quartiles based on different intensities of physical activity (VPA, MPA, walking, or sedentary time) to analyze the distribution of BMI, BMI z-score and weight status. Spearman’s correlations were calculated between different intensities of physical activity.

Multiple linear regression models and nominal logistic regression models were employed to investigate the prospective association of VPA, MPA, walking, and sedentary time with variables of weight status changes (changes in BMI, BMI z-score, and weight status). Time spent in each intensity of physical activity was divided by 30, converted to units of 30 min. “Total time” was also created, meaning the sum of all intensities of physical activity. Three different types of analyses were applied. The first analysis was a single-factor model that examined the association of each intensity category with weight status changes, including age, sex, and province as covariables. The second analysis was a partition model that examined the association of each intensity category while controlling other intensities of activity, with adjustment for covariates. The third analysis involved an isotemporal substitution model that examined the association of substituting 30 min/day of each intensity category with an equal amount of time spent in the other intensity, and the total time of all intensity activities was kept in the equation. In this part, two models were employed. Model 1 was adjusted for the child’s age, sex, and province. Model 2 was further adjusted for parental educational levels, monthly family income, urban/rural residence, history of family obesity, sleep duration, sugary beverage intake, snack intake, and fried food intake. When more than one independent variable was in the model, we calculated a variance inflation factor (VIF) for each variable to evaluate multicollinearity with the cutoff of <4.0. A two-sided *p* < 0.05 was set to assess statistical significance. All the data were analyzed with SPSS 25.0 (SPSS Inc., Chicago, IL, USA).

## 3. Result

### 3.1. Sample Description

Table 1 describes the 15,100 participants (46.7% boys) with characteristics at baseline divided by sex. On average, children were 11.7 years old. No statistically significant differences were found between boys and girls in location and household income, but girls tended to have parents with higher educational levels. Compared to the girls, boys were characterized by a higher proportion of sugary beverages intake and snack intake. Besides, girls spend less time on VPA, MPA and walking, and more time sedentary than boys. About 31.5% of participants had MVPA greater than 60 min per day. Based on age- and sex- specific Chinese criteria of weight status, 2.9% of the sample became overweight/obese at follow-up, and 4.0% became underweight.

Low correlations were observed between different intensities of physical activity, although significant positive associations were found between VPA and MPA, between VPA and walking, and between MPA and walking. The highest observed mean VIF was 1.321, indicating that multicollinearity was not present (Table 2).

### 3.2. Distribution of Changes in BMI, BMI Z-Score and Weight Status According to Quartiles of Different Intensities of Physical Activity

As shown in Table 3, increasing amounts of VPA were associated with greater BMI z-score at baseline (*p* < 0.001), and a similar trend was found in MPA with marginal significance (*p* for trend = 0.072). At the meantime, more time spent in walking was associated with increased BMI at baseline (*p* < 0.001) as well as increased changes in both BMI (*p* = 0.021) and BMI z-score (*p* < 0.001) after a nine-month follow-up. The same results were also found in sedentary time (*p* < 0.001). Moreover, it is shown that more time spent walking or being sedentary was associated with a lower incident of being underweight.

### 3.3. Association between Different Intensity Categories and Changes in both BMI and BMI Z-Score

As listed in Table 4, higher MPA was associated with a higher increase in BMI (*p* = 0.011) and BMI z-score (*p* = 0.026) in the single-factor model but not in the partition model. Walking was also associated with a higher BMI z-score in the single-factor model, but not in the partition model. As for the isotemporal substitution model, reassigning time from sedentary time to VPA or MPA was not associated with changes in BMI and BMI z-score. Replacing 30 min/d ST with an equal duration of walking yielded a small but significantly relative higher BMI z-score (Model 1: *B* = −0.011, *p* = 0.045). However, after adjustment for monthly family income, parental educational level, living in an urban/rural area, history of family obesity, sleep duration, sugary beverage intake, high-energy snack intake, and fried food intake, this association became insignificant.

### 3.4. Association between Different Intensity Categories and Changes in Weight Status

Greater VPA was associated with less risk of overweight/obesity in single-factor model (*OR* = 0.868, *p* = 0.031). In the partition model, adding 30 min/d VPA while holding other categories constant was associated with a 15.6% reduction of overweight/obesity risk (*OR* = 0.844, *p* = 0.026). In the substitution model, replacing 30 min/d of sedentary time with an equal amount time of VPA was associated with a 16.1% decrease in risk of overweight/obesity, which remained statistically significant after adjusting for covariates (*OR* = 0.839, *p* = 0.022). No other substitution effects were found among the other intensity categories. Besides, greater VPA, MPA, and walking were also associated with less risk of being underweight in the single-factor model. However, these results were insignificant when adjusting for the other intensity categories in the partition model. No substitution association with the risk of being underweight was found (Table 5).

## 4. Discussion

By using a population-based sample of Chinese children, the present study applied isotemporal substitution paradigm to examine time reallocation effect of different intensity-specific physical activity on weight status. To the best of our knowledge, this prospective study is the first to investigate the substitution effect of physical activity on incident overweight/obesity. The results suggested that reallocating 30 min/d sedentary time to the equal duration of VPA was favorably associated with a 16.1% reduced risk of incident overweight/obesity. However, the findings showed no beneficial effect of reallocating time of VPA into MPA or walking. A nonsignificant result was also found in replacing sedentary time with walking or MPA.

Normal-weight children with higher VPA had a reduced risk of overweight/obesity. In the present study, the nine-month cumulative incidence of overweight/obesity among Chinese children aged 7 to 18 years was approximately 2.9% during 2013–2014, which was consistent with the substantial increasing prevalence of children. This study found an inverse association between VPA and risk of incident overweight/obesity both in the single and partition model. Numerous previous studies paid attention to this independent association with childhood obesity showing consistent results [21]. A systematic review concluded a negative association between level of physical activity and obesity in children and adolescents [22]. A study conducted among Canadian children aged 7–19 years reported that only VPA was consistently associated with lower waist circumference (WC) and BMI z-score [23].

Isotemporal substitution models in our study further revealed the effect of VPA, showing that replacing sedentary time with an equal duration of VPA was associated with a 16.1% decreased risk of childhood obesity. Previous studies also showed inverse association between replacing sedentary time with VPA and childhood obesity. K. Dalene et al. reported that replacing 10 min/day sedentary time with VPA and MPA were cross-sectionally associated with lower BMI and WC both in children and adolescents [24]. Other cross-sectional studies indicated that replacing 10 min/day of sedentary time with VPA was associated with fat mass index among children aged 6–11 years [25], and substituting 5 min/day sedentary time with VPA was associated with adiposity markers among preschool children [26]. Our prospective results supported research into the effect of VPA on childhood obesity. A system review showed that both reallocating time to VPA and MPA from sedentary was associated with improving adiposity in youth, but the magnitude of association was largest for reallocating VPA [14]. Accordingly, VPA seems to be the most beneficial for obesity prevention based on time allocation with other activity behavior. Biologically, arguments explaining why activity intensity may be relevant to obesity are speculative, including appetite regulation and insulin sensitivity improvement [27]. Evidence from an intervention study suggests that high-intensity physical activity as aerobic exercise can increase subcutaneous adipose nutritive blood flow and reduce suppression of lipolysis [28]. There is increasing emphasis that VPA can incite improvements in body composition and obesity. Taking together with our findings, shifting time from sedentary to VPA in the finite 24-h day can be an efficient first-order public health intervention strategy for reducing incident childhood obesity.

No beneficial effects of replacing MPA or walking with VPA on reducing obesity were found in our study. A study among Finland children aged four years reported no significant association of BMI, percentage of body fat, and fat mass index (FMI) when replacing 5 min per day of MPA with VPA. However, significantly substitution associations with free-fat mass index (FFMI) were observed [26]. Results in adults were also found, suggesting that replacing 1 h per day of LPA with VPA was associated with lower waist circumference, abdominal fat, and visceral fat [29]. Overall, replacement of MPA or others with VPA was significantly beneficial for obesity, which is different from our results. This discrepancy may result from the methodological difference in confounders adjustment and the sample difference in ages. Our finding that replacing walking with VPA was no longer associated with decreased risk of overweight/obesity after adjustment for lifestyle behaviors, such as high-energy snack intake, fried foods intake, and sleep duration. These confounders may influence the association between physical activity and obesity, which often have not been considered in past studies [26,30]. Besides, the diversities of growth status may contribute to the different patterns of physical activity. Most children follow a pattern of a marked decline in MVPA from childhood to young adulthood [9,30], and the annual rate of decline in VPA was the greatest of any other observed changes [31]. It can be concluded that adults had a lower proportion of MVPA in a 24 h day than children, especially lower in VPA, which may go some way to explain the inconsistent results of sample in different growth status.

No association was observed in this study of replacing lower intensity activity with MPA on childhood obesity. However, a cross-sectional study in Finland has shown negative associations between MPA and FMI as well as between reallocating MPA into sedentary and FMI among children aged 6–8 years [25]. A study in Norway also reported similarly inverse association of reallocating MPA into sedentary with BMI and WC among children aged 9 and aged 15 in cross-sectional analysis, but not in prospective analysis [24]. In our prospective study, no beneficial effects of reallocating MPA into sedentary time were found. The substitution effect of MPA to sedentary time on children’s obesity has been seldom reported in prospective studies, while replacing sedentary time with MVPA was estimated to be associated with BMI or body fat [32]. Only few studies have modeled the substitution effect of sedentary time with MPA and VPA separately in children and adolescents (not merged into MVPA); our findings extend previous observations. A plausible explanation for not finding a substitution effect of MPA to lower intensity activity may be related to the study population. Except for VPA, the health benefit of other lower-intensities of physical activity (such as MPA, LPA) among children still unclear [33]. The positive effects of MPA or LPA would be more pronounced in an inactive population [34] and children at-risk of being overweight or obese [35,36]. In the current study, we only included the normal-weight children aged 7 to 18 years at baseline, and nine-month cumulative incidence of overweight/obesity was 2.9%. Additionally, 31.5% of participants adhered the MVPA guideline of 60 min per day. It means that children in our research are more active and with healthier weight status than in other studies.

The present study found a positive association between VPA, MPA, or walking and a reduced risk of being underweight in the single-factor model, but no association was found between being sedentary and underweight. Studies were scarce in examining the association between physical activity and being underweight. A recent study of Polish adolescents reported that underweight boys were characterized by less time spent in MVPA [37]. MVPA might induce improvement in skeletal muscle mass and fiber diameter [38]. Results from another study indicated that VPA or MVPA was prospectively associated with higher FFMI, but not with FMI and fat mass [39]. In this context, we speculate that VPA and MPA may lead to changes in free-fat mass rather than fat mass, resulting in improving weight status and reducing underweight. Although we found no substitution effect of PA on being underweight, our finding adds to the existing knowledge suggesting potential health benefits of physical activity on reducing underweight.

Existing results from cumulative literature and our findings call for urgent action for policy to promote physical activity and improve adiposity [40]. Currently, much sedentary time occurs at school for children and adolescents. Therefore, replacing sedentary time with VPA at school as a strategy is warranted. Moreover, promoting vigorous-intensity physical activity bouts during physical education classes could also be encouraged [41]. According to the guideline for physical activity, policies and programs among school-aged children need to be planned and developed with public support from local government, community, and family.

The main strengths of this study included prospective design with a large sample size and objectively measurement of height and weight, resulting in estimating incidence of overweight/obesity. However, the limitations of this study warrant mentioning. Firstly, different from the objectively measured device, we used the IPAQ to evaluate PA, which may be prone to measurement error and lead to recall bias. However, acceptable reliability and validity of this questionnaire were found, as is mentioned above. Secondly, we only analyzed the data of physical activity at baseline, and did not check the change in daily PA. Thirdly, the nine-month follow-up period maybe not long enough to estimate the incidence of overweight/obesity accurately. Nonetheless, it is still valuable for understanding the association between PA and childhood obesity. Lastly, BMI is just one outcome of obesity. To further clarify the direction of the association between physical activity and childhood obesity, a reasonable follow-up period and other metabolic outcomes should be considered in future longitudinal research.

## 5. Conclusions

This study found that VPA was beneficial for obesity, and replacing 30 min/day of sedentary time with an equal amount of VPA was prospectively associated with a reduced risk of incident overweight/obesity among normal-weight children in China. VPA will provide the best time investment returns for both overweight and underweight children. Our observations suggest that efforts should strive to promote VPA while reducing sedentary time among school-aged children. 

## Figures and Tables

**Table 1 ijerph-17-05761-t001:** General characteristics of children at baseline by sex.

	Total	Boys	Girls	*p* Value
	(*n* = 15,100)	(*n* = 7057)	(*n* = 8043)	
Age at baseline, mean (SD)	11.72 (3.1)	11.59 (3.1)	11.83 (3.1)	<0.001
BMI at baseline, mean (SD)	17.62 (2.2)	17.56 (2.1)	17.67 (2.4)	0.002
BMI z-score at baseline, mean (SD)	−0.15 (0.7)	−0.09 (0.7)	−0.20 (0.7)	<0.001
Residence, *n* (%)				0.408
Urban	9528 (63.1%)	4866 (62.7%)	5589 (63.2%)	
Rural	5572 (36.9%)	2890 (37.3%)	3260 (36.8%)	
Parental educational level, *n* (%)				0.028
Junior high school or below	6635 (48.8%)	3146 (50.2%)	3489 (47.7%)	
Senior high school	3692 (27.2%)	1681 (26.8%)	2011 (27.5%)	
Junior college	1712 (12.6%)	764 (12.2%)	948 (13.0%)	
College or above	1545 (11.4%)	681 (10.9%)	864 (11.8%)	
Monthly family income, *n* (%)				0.74
<5000 RMB	4547 (35.5%)	2067 (35.2%)	2480 (35.7%)	
5000–7999 RMB	2348 (18.3%)	1058 (18.0%)	1290 (18.5%)	
≥8000 RMB	2306 (18.0%)	1060 (18.1%)	1246 (17.9%)	
Refuse to answer	3619 (28.2%)	1679 (28.6%)	1940 (27.9%)	
Sugary Beverages intake, *n* (%)				<0.001
Yes	10073 (67.7%)	4947 (71.1%)	5126 (64.6%)	
No	4811 (32.3%)	2006 (28.9%)	2805 (35.4%)	
High-energy snacks intake, *n* (%)				<0.001
Yes	11225 (75.4%)	5084 (73.1%)	6141 (77.4%)	
No	3659 (24.6%)	1869 (26.9%)	1790 (22.6%)	
Fried food intake, *n* (%)				0.782
Yes	8699 (58.4%)	4072 (58.6%)	4627 (58.3%)	
No	6185 (41.6%)	2881 (41.4%)	3304 (41.7%)	
History of family obesity, *n* (%)				
Yes	2385 (18.7%)	1057 (18.1%)	1328 (19.3%)	0.093
No	10337 (81.3%)	4778 (81.9%)	5559 (80.7%)	
Sleep duration in 9–11 h, *n* (%)				0.051
Yes	3179 (22.4%)	1514 (23.1%)	1665 (21.8%)	
No	11005 (77.6%)	5026 (76.9%)	5979 (78.2%)	
VPA (min/day), mean (SD)	27.03 (35.0)	31.77 (38.4)	22.88 (31.0)	<0.001
MPA (min/day), mean (SD)	26.16 (34.4)	28.73 (36.7)	23.91 (32.1)	<0.001
Walking (min/day), mean (SD)	42.04 (47.0)	44.08 (49.0)	40.25 (45.0)	<0.001
Sedentary (min/day), mean (SD)	338.93 (220.3)	327.27 (219.6)	349.05 (220.4)	<0.001
Weight status at follow-up, *n* (%)				<0.001
Underweight	599 (4.0%)	361 (5.2%)	238 (3.0%)	
Normal	13902 (93.1%)	6405 (91.9%)	7497 (94.2%)	
Overweight or Obesity	434 (2.9%)	207 (3.0%)	227 (2.9%)	

Note: Data were presented as mean (SD) or *n* (%); BMI—body mass index, VPA—vigorous-intensity physical activity, MPA—moderate-intensity.

**Table 2 ijerph-17-05761-t002:** Correlation for different intensity categories and multicollinearity.

	VPA	MPA	Walking	Sedentary	VIF
VPA	1				1.273
MPA	0.457 **	1			1.321
Walking	0.208 **	0.280 **	1		1.095
Sedentary	−0.076 **	−0.070 **	0.007	1	1.009

** Significant at *p* < 0.001, VPA—vigorous-intensity physical activity, MPA—moderate-intensity, VIF—variance inflation factor.

**Table 3 ijerph-17-05761-t003:** Distribution of changes in BMI, BMI z-score and weight status according to quartiles of different intensity categories.

Activity at Baseline	Median ^a^	BMI at Baseline	BMI Z-Score at Baseline	Change in BMI	Change in BMI Z-score	Underweight in Follow-Up	Overweight/Obesity in Follow-Up
	(min/day)	(kg/m^2^)		(kg/m^2^)		(%)	(%)
**VPA**							
Quartile 1	0	17.393	−0.173	0.128	−0.103	4.19%	2.54%
Quartile 2	8.57	17.817	−0.17	0.163	−0.093	4.40%	3.12%
Quartile 3	25	17.716	−0.155	0.152	−0.102	3.92%	2.99%
Quartile 4	60	17.612	−0.103	0.156	−0.106	3.43%	2.82%
*p*		<0.001 **	<0.001 **	0.271	0.529	0.186	0.45
*p* for trend		0.06	<0.001 **	0.337	0.436	0.040*	0.78
**MPA**							
Quartile 1	0	17.787	−0.151	0.15	−0.1	4.06%	3.30%
Quartile 2	8.57	17.7	−0.169	0.143	−0.1	4.11%	2.31%
Quartile 3	22.86	17.595	−0.15	0.148	−0.102	3.92%	3.00%
Quartile 4	60	17.436	−0.131	0.168	−0.096	3.60%	2.94%
*p*		<0.001 **	0.171	0.618	0.946	0.696	0.082
*p* for trend		<0.001 **	0.072	0.239	0.685	0.242	0.97
**Walking**							
Quartile 1	0	17.591	−0.125	0.12	−0.118	5.01%	2.77%
Quartile 2	15	17.453	−0.144	0.154	−0.099	3.67%	3.17%
Quartile 3	40	17.69	−0.17	0.156	−0.095	3.58%	2.76%
Quartile 4	120	17.692	−0.16	0.18	−0.086	3.38%	2.84%
*p*		<0.001 **	0.077	0.021 *	0.005 *	0.002 *	0.674
*p* for trend		<0.001 **	0.132	0.009 *	0.003 *	0.013 *	0.8
**Sedentary**							
Quartile 1	60	17.177	−0.138	0.089	−0.128	4.44%	2.44%
Quartile 2	240	17.205	−0.139	0.144	−0.104	4.53%	3.04%
Quartile 3	440	17.682	−0.158	0.157	−0.094	3.76%	2.99%
Quartile 4	600	18.408	−0.16	0.195	−0.08	3.23%	3.17%
*p*		<0.001 **	0.422	<0.001 **	<0.001 **	0.019 *	0.288
*p* for trend		<0.001 **	0.116	<0.001 **	<0.001 **	0.003 *	0.096

^a^ Median in each quantile group ***** Significant at *p* < 0.05, ****** Significant at *p* < 0.001. VPA—vigorous-intensity physical activity, MPA—moderate-intensity.

**Table 4 ijerph-17-05761-t004:** Single-factor, partition, and isotemporal substitution models of association between 30 min/d change in different intensity categories and changes in BMI and BMI z-score.

	VPA	MPA	Walking	Sedentary
	*B* (95% CI)	*B* (95% CI)	*B* (95% CI)	*B* (95% CI)
Change in BMI				
Single-factor model ^a^	0.005 (−0.008, 0.017)	0.018 (0.005, 0.031) *	0.008 (−0.001, 0.018) ^#^	0.000 (−0.002, 0.002)
Partition model ^b^	−0.004 (−0.019, 0.011)	0.015 (−0.001, 0.030) ^#^	0.005 (−0.006, 0.015)	0.000 (−0.003, 0.002)
Isotemporal substitution model				
Replace VPA				
Model 1	Drop	0.005 (−0.019, 0.029)	0.006 (−0.011, 0.024)	0.000 (−0.014, 0.014)
Model 2	Drop	0.019 (−0.007, 0.044)	0.009 (−0.011, 0.028)	0.004 (−0.011, 0.019)
Replace MPA				
Model 1	−0.005 (−0.029, 0.019)	Drop	0.002 (−0.017, 0.020)	−0.005 (−0.020, 0.009)
Model 2	−0.019 (−0.044, 0.007)	Drop	−0.010 (−0.031, 0.010)	−0.015 (−0.031, 0.001) ^#^
Replace walking				
Model 1	−0.006 (−0.024, 0.011)	−0.002 (−0.020, 0.017)	Drop	−0.007 (−0.017, 0.003)
Model 2	−0.009 (−0.028, 0.011)	0.010 (−0.010, 0.031)	Drop	−0.005 (−0.016, 0.006)
Replace Sedentary				
Model 1	0.000 (−0.014, 0.014)	0.005 (−0.009, 0.020)	0.007 (−0.003, 0.017)	Drop
Model 2	−0.004 (−0.019, 0.011)	0.015 (−0.001, 0.031) ^#^	0.005 (−0.006, 0.016)	Drop
Change in BMI z-score				
Single-factor model ^a^	0.001 (−0.006, 0.007)	0.008 (0.001, 0.014) *	0.005 (0.000, 0.009) *	0.000 (−0.001, 0.001)
Partition model ^b^	−0.003 (−0.011, 0.004)	0.006 (−0.001, 0.014) ^#^	0.004 (−0.001, 0.009)	0.000 (−0.001, 0.001)
Isotemporal substitution model				
Replace VPA				
Model 1	Drop	0.005 (−0.007, 0.016)	0.008 (−0.001, 0.016) ^#^	0.003 (−0.004, 0.009)
Model 2	Drop	0.010 (−0.003, 0.022)	0.007 (−0.002, 0.016)	0.003 (−0.004, 0.011)
Replace MPA				
Model 1	−0.005 (−0.016, 0.007)	Drop	0.003 (−0.006, 0.012)	−0.002 (−0.009, 0.005)
Model 2	−0.010 (−0.022, 0.003)	Drop	−0.003 (−0.013, 0.007)	−0.007 (−0.014, 0.001) ^#^
Replace walking				
Model 1	−0.008 (−0.016, 0.001) ^#^	−0.003 (−0.012, 0.006)	Drop	−0.005 (−0.010, −0.001) *
Model 2	−0.007 (−0.016, 0.002)	0.003 (−0.007, 0.013)	Drop	−0.004 (−0.009, 0.001)
Replace Sedentary				
Model 1	−0.003 (−0.009, 0.004)	0.002 (−0.005, 0.009)	0.005 (0.001, 0.010) *	Drop
Model 2	−0.003 (−0.011, 0.004)	0.007 (−0.001, 0.014) ^#^	0.004 (−0.001, 0.009)	Drop

Note: **^#^** Significant at *p* < 0.1, ***** Significant at *p* < 0.05. ^a^ Single-factor model results were adjusted for age, sex, province, monthly family income, parental educational level, living in an urban/rural area, history of family obesity, sleep duration, sugary beverage intake, high-energy snack intake, and fried food intake. ^b^ Partition model results were adjusted for other intensity categories, age, sex, province, monthly family income, parental education level achieved, living in urban/rural area, history of family obesity, sleep duration, sugary beverage intake, high-energy snack intake, and fried food intake. Model 1: Results were adjusted for age, sex, and province. Model 2: Results were further adjusted for monthly family income, parental educational level, living in urban/rural area, history of family obesity, sleep duration, sugary beverage intake, high-energy snack intake, and fried food intake. BMI—body mass index, VPA—vigorous-intensity physical activity, MPA moderate-intensity.

**Table 5 ijerph-17-05761-t005:** Single-factor, partition, and isotemporal substitution models of association between 30 min/d change in different intensity categories and changes in weight status.

	VPA	MPA	Walking	Sedentary
	*OR* (95% CI)	*OR* (95% CI)	*OR* (95% CI)	*OR* (95% CI)
Overweight and obesity				
Single-factor model ^a^	0.868 (0.763, 0.987) *****	1.017 (0.907, 1.141)	1.011 (0.925, 1.104)	1.007 (0.986, 1.027)
Partition model ^b^	0.844 (0.728, 0.980) *****	1.046 (0.910, 1.202)	1.010 (0.915, 1.116)	1.006 (0.986, 1.027)
Isotemporal substitution model				
Replace VPA				
Model 1	Drop	1.133 (0.936, 1.370)	1.177 (1.018, 1.361) *****	1.133 (1.010, 1.272) *****
Model 2	Drop	1.239 (0.975, 1.573) #	1.197 (0.991, 1.444) #	1.192 (1.026, 1.384) *****
Replace MPA				
Model 1	0.883 (0.730, 1.068)	Drop	1.039 (0.894, 1.208)	1.001 (0.894, 1.121)
Model 2	0.807 (0.636, 1.026) #	Drop	0.966 (0.798, 1.169)	0.962 (0.837, 1.106)
Replace walking				
Model 1	0.850 (0.735, 0.982) *****	0.962 (0.828, 1.118)	Drop	0.963 (0.890, 1.042)
Model 2	0.836 (0.692, 1.009) #	1.035 (0.855, 1.253)	Drop	0.996 (0.899, 1.103)
Replace Sedentary				
Model 1	0.882 (0.786, 0.990) *****	0.999 (0.892, 1.119)	1.038 (0.960, 1.123)	Drop
Model 2	0.839 (0.723, 0.974) *****	1.039 (0.904, 1.195)	1.004 (0.907, 1.112)	Drop
Underweight				
Single-factor model ^a^	0.876 (0.777, 0.988) *****	0.847 (0.748, 0.958) *****	0.900 (0.829, 0.978) *****	1.002 (0.983, 1.021)
Partition model ^b^	0.909 (0.789, 1.047)	0.961 (0.841, 1.098)	0.923 (0.845, 1.009) #	1.004 (0.985, 1.023)
Isotemporal substitution model				
Replace VPA				
Model 1	Drop	1.097 (0.913, 1.318)	1.052 (0.919, 1.204)	1.101 (0.985, 1.231) #
Model 2	Drop	1.058 (0.839, 1.333)	1.016 (0.855, 1.208)	1.105 (0.959, 1.274)
Replace MPA				
Model 1	0.911 (0.759, 1.095)	Drop	0.959 (0.834, 1.102)	1.004 (0.901, 1.118)
Model 2	0.946 (0.750, 1.192)	Drop	0.961 (0.807, 1.143)	1.045 (0.913, 1.196)
Replace walking				
Model 1	0.951 (0.830, 1.089)	1.043 (0.908, 1.199)	Drop	1.047 (0.975, 1.125)
Model 2	0.984 (0.828, 1.170)	1.041 (0.875, 1.239)	Drop	1.088 (0.992, 1.192) #
Replace Sedentary				
Model 1	0.908 (0.812, 1.015) #	0.996 (0.894, 1.110)	0.955 (0.889, 1.026)	Drop
Model 2	0.905 (0.785, 1.043)	0.957 (0.836, 1.095)	0.919 (0.839, 1.008) #	Drop

**Note:**^#^ Significant at *p* < 0.1, ***** Significant at *p* < 0.05. ^a^ Single-factor model results were adjusted for age, sex, BMI z-score at baseline, province, monthly family income, parental educational level, living in an urban/rural area, history of family obesity, sleep duration, sugary beverage intake, high-energy snack intake, and fried food intake. ^b^ Partition model results were adjusted for other intensity categories, age, sex, BMI z-score at baseline, province, monthly family income, parental educational level, living in an urban/rural area, history of family obesity, sleep duration, sugary beverage intake, high-energy snack intake, and fried food intake. Model 1: Results were adjusted for age, sex, BMI z-score at baseline and province. Model 2: Results were further adjusted for monthly family income, parental educational level, living in an urban/rural area, history of family obesity, sleep duration, sugary beverage intake, high-energy snack intake, and fried food intake. VPA—vigorous-intensity physical activity, MPA—moderate-intensity.

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
