# Peer review of "Association of Reallocating Time in Different Intensities of Physical Activity with Weight Status Changes among Normal-Weight Chinese Children: A National Prospective Study"

_ijerph, 2020, doi:10.3390/ijerph17165761_

Round 1

Reviewer 1 Report

This manuscript submitted by Tan et al entitled as “Association of Reallocating Time in Different Intensity of Physical Activities with Weight Status Changes Among Normal-weight Chinese Children: A National Prospective Study” attempted to investigate prospective association between reallocating time in different intensity of physical activity and weight status changes among Chinese children. Author demonstrated that each 30 min/day of increase in VPA was favorably associated with a reduced risk of incident overweight/obesity.  Author concluded that there is need for promoting vigorous-intensity physical activity in children.  Below are some concerns related to the study as:

Comment 1:Why author didn’t found association in replacing MPA to lower intensity with childhood obesity, please explain in discussion section.

Comment 2:In the follow-up study, did author found underweight childrens, if yes what was the percentage. 

Comment 3:What was the relationship between family status with the on set of obesity.

Comment 4: Why author selected 9 months for follow up. Any reasons?

Comment 5:In discussion section second last paragraph it should be follow-up NOT followerup.

Comment 6:Why author didn’t recruit obese students for better comparisons with normal weight students.

Author Response

We would like to express our greatest appreciation to the reviewer for the great comments. Those comments are valuable and very helpful for revising and improving our paper, as well as the important guiding significance to our researchers. We have revised our manuscript according to these comments. We believe that the quality of our manuscript has been improved significantly. Below please find our point by point responses to these comments.

Reviewer #1:

This manuscript submitted by Tan et al entitled as “Association of Reallocating Time in Different Intensity of Physical Activities with Weight Status Changes Among Normal-weight Chinese Children: A National Prospective Study” attempted to investigate prospective association between reallocating time in different intensity of physical activity and weight status changes among Chinese children. Author demonstrated that each 30 min/day of increase in VPA was favorably associated with a reduced risk of incident overweight/obesity.  Author concluded that there is need for promoting vigorous-intensity physical activity in children.  Below are some concerns related to the study as:

Comment 1: Why author didn’t found association in replacing MPA to lower intensity with childhood obesity, please explain in discussion section.

Responses:

Many thanks for your comments. A plausible explanation for not finding association of replacing MPA to lower intensity with childhood obesity may be the sample particularity. The positive effects of MPA or LPA would be more pronounced in inactive population[1] and at-risk overweight or obesity children[2-3]. In the current study, we only included the normal-weight children aged 7 to 18 years at baseline, and 9-month cumulative incidence of overweight/obesity was 2.9%. Additionally, 31.5% of participants adhered the MVPA guideline of 60 min per day. It means that children in our research are more active and with healthier weight status.

We have modified the Discussion of manuscript to explain the possible reasons (Page 11, Line 322-336).

Reference:

[1]Aggio, D.; Smith, L.; Hamer, M., Effects of reallocating time in different activity intensities on health and fitness: a cross sectional study. Int J Behav Nutr Phys Act 2015, 12, 83.

[2]Joseph, R. P.; Casazza, K.; Durant, N. H., The effect of a 3-month moderate-intensity physical activity program on body composition in overweight and obese African American college females. Osteoporos Int 2014, 25, (10), 2485-2491.

[3]Mitchell, J. A.; Pate, R. R.; Beets, M. W.; Nader, P. R., Time spent in sedentary behavior and changes in childhood BMI: a longitudinal study from ages 9 to 15 years. Int J Obes (Lond) 2013, 37, (1), 54-60.

Comment 2: In the follow-up study, did author found underweight children, if yes what was the percentage. 

Responses:

Yes, we also found underweight children in the follow-up study. Based on age- and sex- specific Chinese criteria of weight status, 4.0% of the sample turned into underweight at follow-up. We have mentioned in the Results. (Page 4, Line 231)

Comment 3: What was the relationship between family status with the on-set of obesity.

Responses:

We explored the difference in weight status and monthly family income by  chi-square tests , which presented in Table R1. We found no significant difference in level of family income among different weight status.

Table R1 Difference in weight status and monthly family income

Normal weight

Overweight/obesity

Underweight

P

Monthly family income, n (%)

<5000 RMB

4212 (35.7)

122 (33.2)

158 (31.9)

0.139

5000-7999 RMB

2141 (18.1)

84 (22.9)

91 (18.3)

≥8000 RMB

2136 (18.1)

63 (17.2)

89 (17.9)

Refuse to answer

3325 (28.1)

98 (26.7)

158 (31.9)

Comment 4: Why author selected 9 months for follow up. Any reasons?

Responses:

Our current study is based on a national school-based health lifestyle intervention program from September 2013 to June 2014. And the nine months from September to June in the next year is the typical academic year in China. However, the 9-month follow-up period maybe not long enough to estimate the incidence of overweight/obesity accurately, so we have acknowledged this as a limitation in the Discussion of the manuscript. (Page 12, Line 465-467)

Comment 5: In discussion section second last paragraph it should be follow-up NOT followerup.

Responses:

We are very sorry for our incorrect writing. Now we have corrected the “follower” to “follow”(Page 12, Line 465)

Comment 6: Why author didn’t recruit obese students for better comparisons with normal weight students.

Responses:

Many thanks for your advice. Our current study aims to investigate the prospective association between time reallocation in physical activity and incident obesity, and pays attention to the normal-weight children for obesity prevention. Under this circumstance, we didn’t recruit obese children for comparisons.

Reviewer 2 Report

The Authors prepared an interesting work with high scientific value. Low physical activity and its conseqences are very serious esepecially when it comes to children, adolescents and young adults. The manuscript is well pepared but there are some issues to discuss.

Introduction

The Authors should specify some data.

Line 36-the Authors provide data of 340 milion- without area of incidence, is it a global result?

Line 40, global guidelines – the Authors mentioned only about MVPA recommendations, what about VPA? It should be also specified. You should add the information who is the author of recommendation, now there is only information thatthey are global

Line 56- several helath outcomes- please specify..

Line 63- the Authors write about guildelines for reducing sedentary time in young people-but the study group consists children and adolescents- you should add these groups in this sentence

Methods

Please add the information about % rate of the children who completed both studies (I study and follow –up). Specify the exclusion criteria, do you took into acount the presence of chronić diseases ?

Anthropometric measurement – what kind of specialist did the measurment?

Questionnaire – do the youngest children fill the answers in the questionnaire alone or with the help of parents?

Please add detailed information about metodology in the follow up study? Was it the same? What about measurment of physical activity? Do you checked the change in physical activity? If not please add this issue in the limitations of the study

Table 2-please  specify abbreviation VIF

Author Response

We would like to express our greatest appreciation to the reviewer for the great comments. Those comments are valuable and very helpful for revising and improving our paper, as well as the important guiding significance to our researchers. We have revised our manuscript according to these comments. We believe that the quality of our manuscript has been improved significantly. Below please find our point by point responses to these comments.

Reviewer #2: The Authors prepared an interesting work with high scientific value. Low physical activity and its consequences are very serious especially when it comes to children, adolescents and young adults. The manuscript is well prepared but there are some issues to discuss.

Introduction

The Authors should specify some data.

  1. Line 36-the Authors provide data of 340 million- without area of incidence, is it a global result?

Responses:

Thank you for your careful comments. Yes, it is the data from World Health Organization for global children. We have added “around the world” to make it clear. (Page 1, Line 38)

  1. Line 40, global guidelines – the Authors mentioned only about MVPA recommendations, what about VPA? It should be also specified. You should add the information who is the author of recommendation, now there is only information that they are global.

Responses:

Thank you for the suggestions. We have added the information of VPA and the authors of recommendation as follow: “Accordingly, global guidelines, including from World Health Organization and Department of Health and Human Services in America, recommended 60 min per day of MVPA and 3 times per week of vigorous-intensity physical activity (VPA) for children and adolescents to achieve health benefits, including healthy weight status” (Page 1, Line 41-66).

  1. Line 56- several health outcomes- please specify.

Responses:

As suggested, we have specified the health outcomes and revised the manuscript as “... health outcomes (e.g. cardiovascular health, waist circumference change and weight status)” (Page 2, Line 75).

  1. Line 63- the Authors write about guidelines for reducing sedentary time in young people-but the study group consists children and adolescents- you should add these groups in this sentence

Responses:

Thanks for your comments. We have substituted the “young people” with “children and adolescents” as suggested (Page 2, Line 89).

Methods

  1. Please add the information about % rate of the children who completed both studies (I study and follow –up). Specify the exclusion criteria, do you took into account the presence of chronić diseases ?

Responses:

Thank you for your thoughtful comments. We revised the related sentence as “These children ere enrolled in baseline measurement and 98.9% of them (n=18,108) participated the follow-up data collection.A total of 3,008 students were excluded because of missing data on valid assessment of physical activity.”(Page2, Line 107-109) As for exclusion criteria, we have excluded children who suspected to severe physical disease which may have adverse effects on daily physical activity in the original program. However, we didn’t take chronic disease into account in the current study. Now we added the exclusion criteria in Methods. (Page.2, Line102-103)

  1. Anthropometric measurement – what kind of specialist did the measurement?

Responses:

The anthropometric variables were measured by professional nurses and doctors, according to standard procedures. Now we have added the information to make it clarified in the section of Methods (Page 3, Line 152-153).

  1. Questionnaire – do the youngest children fill the answers in the questionnaire alone or with the help of parents?

Responses:

Parents were instructed to assist their children in completing the children-reported questionnaire. But the children-reported questionnaire of children grade 1-3 were reported by parents. As suggested we have revised our manuscript by adding details. (Page 3, Line 163-167)

  1. Please add detailed information about methodology in the follow up study? Was it the same? What about measurement of physical activity? Do you checked the change in physical activity? If not please add this issue in the limitations of the study

Responses:

Thank you for your thoughtful comments. Yes, we conducted the same assessment in the follow-up study, and physical activity was measured by international physical activity questionnaire both at baseline and follow-up. However, we did not check the change in physical activity, so we have added this as a limitation in the Discussion as “Secondly, we only analyzed the data of physical activity at baseline, which may also affect the assessment and did not check the change in daily PA”. (Page 12, Line 464-465)

  1. Table 2-please specify abbreviation VIF

Responses:

VIF is the abbreviation variance inflation factor. Now we have specified the abbreviation VIF accordingly (Page 5, Line241, Table 2, footnote).